# Extracellular c-di-GMP Plays a Role in Biofilm Formation and Dispersion of *Campylobacter jejuni*

**DOI:** 10.3390/microorganisms10102030

**Published:** 2022-10-14

**Authors:** Bassam A. Elgamoudi, Kirstie S. Starr, Victoria Korolik

**Affiliations:** Institute for Glycomics, Griffith University, Gold Coast, QLD 4222, Australia

**Keywords:** *Campylobacter jejuni*, biofilm, dispersion, c-di-GMP

## Abstract

Cyclic diguanosine monophosphate (c-diGMP) is a ubiquitous second messenger involved in the regulation of many signalling systems in bacteria, including motility and biofilm formation. Recently, it has been reported that c-di-GMP was detected in *C.* *jejuni* DRH212; however, the presence and the role of c-di-GMP in other *C. jejuni* strains are unknown. Here, we investigated extracellular c-di-GMP as an environmental signal that potentially triggers biofilm formation in *C. jejuni* NCTC 11168 using a crystal violet-based assay, motility-based plate assay, RT-PCR and confocal laser scanning microscopy (CLSM). We found that, in presence of extracellular c-di-GMP, the biofilm formation was significantly reduced (>50%) and biofilm dispersion enhanced (up to 60%) with no effect on growth. In addition, the presence of extracellular c-di-GMP promoted chemotactic motility, inhibited the adherence of *C. jejuni* NCTC 11168-O to Caco-2 cells and upregulated the expression of *Cj1198* (*luxS*, encoding quarum sensing pathway component, autoinducer-2), as well as chemotaxis genes *Cj0284c* (*cheA*) and *Cj0448c* (*tlp6*). Unexpectedly, the expression of *Cj0643* (*cbrR*), containing a GGDEF-like domain and recently identified as a potential diguanylate cyclase gene, required for the synthesis of c-di-GMP, was not affected. Our findings suggest that extracellular c-di-GMP could be involved in *C. jejuni* gene regulation, sensing and biofilm dispersion.

## 1. Introduction

*Campylobacter* *jejuni* is the most common foodborne pathogen which commonly causes bacterial diarrheal illness worldwide [1,2]. The common symptoms of *C. jejuni* gastroenteritis include mild or severe bloody diarrhoea, with abdominal pain and high fever, that frequently requires hospitalisation and antimicrobial treatment. *C. jejuni* can also cause post-infection complications such as meningitis, urinary tract infections and immune-mediated neuropathies (i.e., Guillian Barré Syndrome) [3,4]. *Campylobacter* spp. can be detected in water reservoirs, as commensals in the gastrointestinal tract of many animals including birds, and as virulent pathogens in humans. Animal reservoirs, including domestic and wild animals, are an important link in the chain of transmission of infectious pathogens to humans [5,6,7,8]. Contaminated animal food products, poultry, in particular, are considered to be a major source of bacteria causing human campylobacteriosis [9,10]. Moreover, *Campylobacter* spp (i.e., *C. jejuni*) display intrinsic resistance to many antimicrobial agents [11,12]. There is increasing resistance of *Campylobacter* spp. to many frequently prescribed antibiotics against campylobacters (i.e., erythromycin, and fluoroquinolones). Campylobacters are now listed on the WHO priority pathogens list for development of new antibiotics [13,14,15]. These pathogens have several potential survival mechanisms such as stress responses, viable but nonculturable state (VBNC), stationary phase survival mechanisms, and, importantly, biofilm formation [16,17,18,19,20]. *Campylobacter* species have reported to be capable of forming a mono-species biofilm and can integrate into a pre-existing biofilm [21,22,23,24,25]. Biofilms have been found to be involved in the transmission of campylobacter disease through multi-microbial species communities, which can be as part of the normal microbiota in animal intestines [7,26,27]. Consequently, biofilms are becoming recognised as a contributing factor of *C*. *jejuni* transmission through the food chain [8,28]. For example, the role of biofilm formation of *C. jejuni* in antibiotics resistance has been previously demonstrated where antimicrobial resistance gene transferred between *C. jejuni* strains within biofilms [12,15,29].

One of the strategies for combating bacterial biofilms is the anti-biofilm compounds, natural or synthetic, which have antimicrobial and chemo-preventive properties [30,31]. Some of those compounds have anti-quorum sensing properties or the ability to modify cyclic di-GMP (c-di-GMP), a universal bacterial second sensory messenger [32]. Here, we focus on c-di-GMP which regulates many cellular biological processes such as biofilm formation, motility, cell cycle, and virulence factors [33]. Generally, low and high levels of intracellular c-di-GMP regulate the transition between motile (planktonic) and sessile states which contributes to slow growth and control biofilm formation [34,35,36,37,38]. Two enzymes that commonly regulate the intracellular c-di-GMP level, a diguanylate cyclase (DGCs) which synthesizes c-di-GMP by converting two GTP molecules to c-di-GMP, and the phosphodiesterase (PDE) enzyme, that degrades c-di-GMP into 5′-phosphoguanylyl-(3′-5′)-guanosine (pGpG) and/or GMP [39]. c-di-GMP had been implicated in all stages of biofilm formation from attachment and proliferation, including dispersal. It also plays a role in the production of extracellular polymeric substances (EPS), as components involved in biofilm assembly, which mainly composed of polysaccharides (or known as exopolysaccharides), extracellular DNA (eDNA), protein adhesins, and [37,38,40,41,42]. Synthetic c-di-GMP has been shown to have an inhibitory effect on biofilm formation of *Staphylococcus mutans* and methicillin-resistant *Staphylococcus aureus* (MRSA) [43,44,45], whereas adding an extracellular c-di-GMP at high concentrations suppressed the biofilm formation [46]. Those studies suggested that using c-di-GMP in combination with other antimicrobial agents, may be a promising novel approach to treating or inhibiting biofilms [47]. Recently, the c-di-GMP signalling system has been reported in *C. jejuni* DRH212 [48]; however, the presence and the role of c-di-GMP in other *C. jejuni* strains, particularly *C. jejuni* NCTC 11168, are still unknown.

In this work, we demonstrate that c-di-GMP (extracellular c-di-GMP) inhibits and disperses biofilms formed by *C. jejuni* strains, and it appeared to promote chemotactic motility. Furthermore, the presence of c-di-GMP inhibited the adherence of *C. jejuni* NCTC 11168 to Caco-2 cells. Interestingly, RT-PCR showed that the presence of c-di-GMP upregulated *Cj1198* (luxS, a quorum sensing signal (autoinducer-2)), as well as chemotaxis genes *Cj0284c* (CheA) and *Cj0448c* (tlp6). Unexpectedly, the presence of extracellular c-di-GMP has not altered on gene expression of *Cj0643* (cbrR, *Campylobacter* bile resistance regulator). *CbrR* is the only gene that contains a GGDEF domain in *C. jejuni* NCTC 11168, which has been identified as a potential diguanylate cyclase that synthesizes c-di-GMP. Our findings suggest that c-di-GMP could be involved in *C. jejuni* biofilm dispersion.

## 2. Materials and Methods

### 2.1. C. jejuni Strains and Growth Conditions

Bacterial strains used in this study were *C. jejuni* NCTC 11168-O, *C. jejuni* 81116 (courtesy of Prof. D. G. Newell, Central Veterinary Laboratories, Surrey, UK), and *C. jejuni* 81-176 (courtesy of Prof. C. Szymanski, University of Alberta, Edmonton, AB, Canada) [49]. Cells were grown microaerobically (85% N_2_, 10% CO_2_ and 5% O_2_) at 42 °C on Mueller-Hinton agar/broth (MHA/MHB) (Thermo Fisher Scientific Inc., Sydney, Australia) and in Mueller-Hinton broth (MHB), which was supplemented with Trimethoprim (2.5 μg mL^−1^) and Vancomycin (10 μg mL^−1^) (TV) (Sigma Aldrich, St. Louis, MO, USA).

### 2.2. Chemicals Used in This Study

Cyclic diguanylate monophosphate (c-di-GMP) compounds provide by InvivoGen (San Diego, CA, USA).

### 2.3. Biofilm Formation and Dispersion Assays

The assays were performed as previously described [50]. Briefly, overnight cultures of *C. jejuni* strains were diluted to an OD_600_ (optical density at 600 nm) of 0.05, and 2 mL of cell suspension were dispensed into 24-wells plates (Geiner Bio-One, Monroe, NC, USA). Extracellular c-di-GMP (ex-c-di-GMP) at different concentrations (50–400 µM) were added directly to the culture in the wells and incubated at 42 °C under microaerobic conditions for 48 h. For dispersion assay, *C. jejuni* cells were grown as described above, except no ex-c-di-GMP was added. Then, PBS containing the appropriate concentration of ex-c-di-GMP (50–400 µM) was added to the wells and plates were incubated for a further 24 h. Concentration of dispersed cells in the aqueous phase was measured (OD_600_) spectrophotometrically, and the remaining biofilm was assessed by crystal violet (CV) staining. After that, the plates were gently rinsed with water, dried at 55 °C for 30 min and stained using the modified CV staining method as described previously [51]. The percentage of biofilm inhibition and dispersion (%) was calculated as described by [50,52,53]. Data are presented as mean ± standard errors.

### 2.4. Swarm Assay

Cell suspensions of overnight cultures of *C. jejuni* were diluted with MHB to an absorbance of 0.1 at OD_600_. Then, 20 μL aliquots (untreated) and mixing with 200 µM of ex-c-di-GMP were spotted in the centre of swarm plates (0.35% MHA) and incubated at 42 °C under microaerophilic conditions. The plates were photographed at 24 h intervals for up to 96 h.

### 2.5. Effect of Extracellular c-di-GMP on *C. jejuni* Growth

*C. jejuni* cells were grown onto MHA plates supplemented with antibiotics (TV) overnight at 42 °C under microaerobic conditions. These plate cultures were used to inoculate 5 mL MHB-TV starter cultures which were then incubated with shaking (50 rpm) overnight microaerobically. The OD_600_ of each starter culture was measured and the cultures were then diluted accordingly in fresh MHB-TV to give a final OD_600_ of 0.03. Triplicate cultures of 5 mL in 15 mL tubes for untreated (control) and treated (with 200 µM of ex-c-di-GMP) samples were prepared and placed under microaerobic conditions shaking at 50 rpm. The OD_600_ was measured at 0, 2, 6, 10, 12, 18 and 24 h using a spectrophotometer.

### 2.6. Auto-Aggregation Assay

The ability of bacteria to auto-aggregate was assessed as described by [21,48]. Briefly, *C. jejuni* cells were harvested from overnight MHA plates, washed with PBS and resuspended in PBS to OD_600_ 0.8. To determine percentage auto-aggregation, two millilitres of bacterial suspension were placed into glass culture tubes with MHB and incubated at room temperature and monitored the decrease in OD_600_ at different times (0, 2 and 24 h). The auto-aggregation percentage was calculated as follows: 100 × [1 – (OD_final_/OD_initial_)] [54].

### 2.7. Adherence Assay with Cultured Caco-2 Cells In Vitro

Adherence rate was calculated against the size of the inoculum, (~10^9^ bacterial cells per human cell in confluent cell culture) in the presence and absence of ex-c-di-GMP. The assay was performed with Caco-2 cells as previously described [55]. Briefly, Caco-2 cells were grown to confluence in a 24-well plate using the minimal essential medium (MEM, Gibco Laboratories, Grand Island, NY, USA) supplemented with 10% foetal bovine serum. Adherence assay was performed with a starting inoculum of ∼10^9^ bacteria per well. Ex-c-di-GMP competition was pre-incubating *C. jejuni* with 200 µM of ex-c-di-GMP. Experiments were repeated using biological replicates.

### 2.8. RT-PCR Analysis

The RT-PCR was performed as described previously in [50]. Briefly, *C. jejuni* 11168-O cells grown overnight at 42 °C with shaking at 50 rpm, under microaerobic conditions. Cells were centrifuged at 4000 rpm for 15 min to collect the pellets and then suspended in MHB and adjusted OD_600_ to 1 (~3 × 10^9^ cells/mL) and subsequently challenged with 200 µM of ex-c-di-GMP for 3-h. The bacterial survival was confirmed by viable cell counts after 3 h. The cells were centrifugated at 4000 rpm for 15 min and pellets were used for RNA extraction by Rneasy kit according to the manufacturer’s protocol (Bioline, Eveleigh NSW, Australia) [56]. To synthesise cDNA, 0.5 μg of extracted RNA was reverse transcribed via Omniscript^®®^ Reverse TranscriptionKit according to the manufacturer’s protocol (Qiagen, Clayton, VIC, Australia). Real-time cycler conditions were 20 s at 95 °C, 3 s at 95 °C followed by 30 s at 60 °C with 40 cycles. All primers used in the present study are listed in (Appendix A), which have ordered from (Sigma Aldrich, St. Louis, MO, USA). Clone Manager (version 9) (Scientific and Educational Software, Westminster, CO, USA) was used to design primers to amplify 100-bp fragments. Amplification plots and melt curves were analysed by Applied Biosystems 7500 Fast real-time PCR system. Samples were normalised using *gyrA* gene as a housekeeping gene [56,57,58]. All the samples, including controls, were analysed in triplicate and each experiment was biological replicates of each sample. Relative *n*-fold changes in the transcription of the examined genes between the treated and non-treated samples were calculated as previously described by [50].

### 2.9. Confocal Laser Scanning Microscopy

Biofilm preparation was previously described in [50]. *C. jejuni* cells, overnight cultures, were diluted to an OD_600_ of 0.05, and 3 mL of each sample was placed into 6-well plate containing a glass coverslip to form the biofilm (Geiner Bio-One, Monroe, NC USA) as described above in dispersion assay. The plates were then incubated at 42 °C microaerobically for 48 h. Following incubation, MH broth was removed and replaced with PBS containing 200 µM ex-c-di-GMP, and the plates were incubated for a further 3 h and then gently washed. The coverslips were fixed using 5% formaldehyde solution for 1 h at room temperature, gently washed and stained with two fluorescent dyes as follows: (I) Calcofluor White (CFW) (Sigma Aldrich, St. Louis, MO, USA), which labels polysaccharides by binding to N-acetylglucosamine and sialic acid residues, at 25 μM final concentration for 20 min [59,60]; (II) coverslips were then mounted on slides under coverslips in DAPI (Sigma Aldrich, USA), containing 4′,6-diamidino-2-phenylindole (DAPI), which labels DNA [50]. All samples were examined by CLSM (Nikon A1R+, Griffith University, Gold Coast, QLD, Australia), and images were processed using ImageJ analysis software version 1.5i (National Institutes of Health, Bethesda, MD, USA).

### 2.10. Statistical Analysis

The statistical significance of data generated in this study was determined using an unpaired two-tailed Student’s *t*-test, which was carried out in GraphPad Prism (GraphPad Software, San Diego, CA, USA). *p* ≤ 0.05 was considered statistically significant.

## 3. Results

### 3.1. Extracellular c-di-GMP Affects the Motility of *C. jejuni*

The examination of the relationship between c-di-GMP and the swarming motility of wild-type *C. jejuni* 11168-O showed that the presence of 200 µM ex-c-di-GMP reduced the ability of *C. jejuni* 11168-O to swarm on soft agar (Figure 1). Ex-c-di-GMP did not affect the growth of *C. jejuni* 11168-O (Appendix A).

### 3.2. Extracellular c-di-GMP Affects Biofilm Formation by C. jejuni

In order to investigate the effect of ex-c-di-GMP on biofilm formation, different concentrations of ex-c-di-GMP (50–400 µM) were tested for their inhibitory effect on *C. jejuni* biofilm by the inhibition assay. The presence of ex-c-di-GMP had an significant inhibitory effect (*p* < 0.001, Appendix A) on biofilm formation by *C. jejuni* 11168-O in a dose-dependent manner (Figure 2A) where 400 µM reduced biofilm formation by 48% (Figure 2A). Furthermore, the dispersion assay was used to determine the effect on the dispersion of a formed biofilm. Ex-c-di-GMP was able to disrupt existing biofilm, where 400 µM ex-c-di-GMP disrupted formed biofilm to greater than 60% extent (Figure 2B and Appendix A).

### 3.3. Extracellular c-di-GMP Affects Biofilm Formation by Different Campylobacter Strains

In order to determine if ex-c-di-GMP affected *C. jejuni* in a strain-specific manner, *C. jejuni* 11168-O, *C. jejuni* 81-176, and *C. jejuni* 81116, were used to demonstrate general inhibitory effect of ex-c-di-GMP on biofilm formation (Figure 3, Appendix A).

### 3.4. Extracellular c-di-GMP Treatment Effects Auto-Aggregation of C. jejuni

Untreated and cultures treated with ex-c-di-GMP were visually examined for visible cell–cell clumping and aggregation. Next, 200 μM ex-c-di-GMP-treated culture of *C. jejuni* 11168-O displayed significantly (*p* = 0.029 and 0.022, respectively) less auto-aggregation, compared to the untreated culture (Figure 4).

### 3.5. Effect of Extracellular c-di-GMP on Adhesion of C. jejuni to Caco-2 Cell Lines

To understand the effect of ex-c-di-GMP on *C. jejuni* 11168-O adhesion ability, we performed a cell adherence assay using polarised mammalian cultured Caco-2 cells in the presence and absence of 200 µM ex-c-di-GMP. As shown in Figure 5, a significant reduction (*p* < 0.05) was observed in adherence of *C. jejuni* 11168-O cells treated with 200 µM Ex-c-di-GMP, as compared with untreated cells to Caco-2 cells.

### 3.6. Dispersal Effect of Extracellular c-di-GMP on C. jejuni Biofilm Can Be Visualized

In order to observe the dispersal effect of ex-c-di-GMP on formed biofilm of *C. jejuni* 11168-O, the mature biofilm of *C. jejuni* 11168-O was treated with ex-c-di-GMP for 3 h (Figure 6 and Appendix A) and then visualized using CLSM. The biofilm architecture showed monolayers of cells covered with the extracellular matrix, likely to be EPS [61]. The treatment of mature biofilm with ex-c-di-GMP appeared to reduce previously formed structures, and the formation of central void within the biofilm structure typical of the biofilm dispersal could be observed.

### 3.7. Extracellular c-di-GMP Dysregulates Specific Gene Expression

To elucidate the mechanism of action of ex-c-di-GMP, its effect on the expression levels of genes previously shown to be dysregulated during biofilm formation [62,63,64,65,66] was examined. The expression of genes involved in chemotactic motility and quorum sensing in *C.jejuni* 11168-O (*cheA*, *tlp6*, *tlp8* and *luxS*) were examined in the presence and absence of ex-c-di-GMP by RT-PCR. *CbrR* gene as the only gene in the *C. jejuni* genome encoding GGDEF-like motif, reported to be potentially involved in c-di-GMP synthesis [48] was also included. The expression of, *tlp6*, *LuxS* and *CheA* was upregulated in presence of 200 µM ex-c-di-GMP by 1.9 to 3-fold (Figure 7); however, no significant effect on the expression of *tlp8* was observed. Surprisingly, no changes in gene expression of *cbrR* have been noted.

## 4. Discussion

c-di-GMP signalling system is involved in the regulation of biofilm development stages from attachment to dispersal, where cells revert from the sessile lifestyle to planktonic and vice versa [26,38,67,68]. Interestingly, introduction of extracellular c-di-GMP (ex-c-di-GMP) has been shown to inhibit biofilm formation in some bacterial species including *S.*
*aureus* and *S. mutans* [43,44,46,47]. Here, we demonstrate that extraneous c-di-GMP can reduce the formation *C. jejuni* biofilms as well as disperse formed biofilms in a dose-dependent manner. Ex-c-di-GMP at concentrations of ≥50 µM was able to significantly trigger the disassembly of the formed biofilms. This is in agreement with previously reported observations of significant inhibitory effect of ex-c-di-GMP on biofilm formation by *S. aureus* and *S. mutans* [44,46,47].

Since our data show that ex-c-di-GMP treatment inhibited *C. jejuni* biofilms, we hypothesised that ex-c-di-GMP may also affect established biofilms. Indeed, ex-c-di-GMP treatment of formed *C. jejuni* biofilms led to a reduction of biofilm development by up to 60. Moreover, while the inhibitory effect of ex-c-di-GMP was observed for all strains tested, the extent of inhibition was strain-dependent. Furthermore, microscopic analysis showed that when the formed biofilm was treated with ex-c-di-GMP, formation of central voids within the biofilm could be observed. This is similar to the reports of void formation with hollow cavities, containing swimming bacterial cells (seeding dispersal cells), within *P. aeruginosa* biofilm during biofilm dispersion [69,70,71].

However, it is still unclear how the ex-c-di-GMP acts to inhibit *C. jejuni* biofilm or how it enters *C. jejuni* cells. One of the possibilities is that *C. jejuni* senses this external compound as a part of the quorum sensing pathway where it might trigger a signalling cascade, modulating gene expression. Here, we have shown that *luxS*, a quorum sensing pathway gene, present in many bacterial species and responsible for the production of autoinducer-2 (AI-2) [72,73,74,75,76], was upregulated in *C. jejuni* in presence of ex-c-di-GMP. Moreover, AI-2 can mediate auto-aggregation and has been linked to pathogenicity, drug resistance, and host immune evasion [77]. Auto-aggregation has been identified as one of the first steps of biofilm formation (cell–cell attachment) [77,78]. This is generally mediated by surface proteins or carbohydrates, particularly exopolysaccharides (i.e., poly-N-acetylglucosamine), which can act as auto-agglutinins [79]. We found that *C. jejuni* cells treated with 200 μM ex-c-di-GMP show reduced aggregation, similar to that reported for *S. aureus* [46].

AI-2 had also been shown to influence chemotaxis and swarming motility [80,81,82]. Here, we found that chemotaxis associated genes *cheA* and *tlp6* are upregulated in presence of ex-c-di-GMP, in agreement with previous report where these genes were shown to be upregulated within formed biofilms [83]. Changes in levels of CheA, in particular, are not surprising as it ultimately controls the directional rotation of the flagellar motor by changing its own phosphorylation levels, and thereby triggering changes in response regulators in the chemotaxis pathway [84]. Motility and flagellar gene expression in *C. jejuni* are indeed affected by *luxS* mutation Jeon et al. [85], where the mutation reduces expression of flagellin gene, *flaA* by 43% leading to reduction in motility. Here, we have shown that addition of ex-c-di-GMP affected *C. jejuni* swarming motility and enhanced the swarm zone achieved by the wild type cells, potentially as the result of overexpression of *luxS*. The change in motility in presence of ex-c-di-GMP could potentially reflect the transition between planktonic and biofilm states.

The ability of c-di-GMP to enter the *Campylobacter* cells can be reasonably postulated, as it has been reported to be able to enter eukaryotic cells [86]. Moreover, some microorganisms are known to secrete c-di-AMP or c-di-GMP as signalling molecules, such as *Bacillus subtilis*, *Listeria monocytogenes* and amoeba *Dictyostelium discoideum* [87,88,89]. For example, *B. subtilis* is able to secret c-di-AMP which impacts biofilm formation and plant root colonization [35,87].

Such messengers can also be detected by the innate immune system and can activate the expression of interferon genes and thus act as pathogen-associated molecular patterns (PAMPs) during infection [90,91]. Indeed, ex-c-di-GMP had been suggested to be a novel immunostimulatory agent that can modulate the host immune response, as was demonstrated by its ability to inhibit *S. aureus* infection in a mouse model [44]. This is consistent with our data, showing adherence of *C. jejuni* cells to the Caco-2 epithelial cells is significantly decreased after treatment with 200 μM ex-c-di-GMP, further suggesting that ex-c-di-GMP can potentially inhibit biofilm formation of biotic surfaces. However, the molecular bases for ex c-di-GMP inhibition of *C. jejuni* epithelial cell adherence are yet to be elucidated.

It is also possible that ex-c-di-GMP enters the cells by binding to the diguanylate cyclase (DGCs) which synthesizes c-di-GMP. Recently, the c-di-GMP signalling system has been reported in *C. jejuni* DRH212 [48], where *cbrR* gene was proposed to be a negative regulator of FlaA expression and motility. However, the presence and the role of c-di-GMP in other *C. jejuni* strains are still unknown, and comparative gene expression between biofilm and planktonic of *C. jejuni* 11168-O cells showed no differences in *cbrR* expression [66]. Similarly, no differences in expression of *cbrR* expression before and after treatment with ex-c-di-GMP have been observed in this study. It is possible, however, that this gene is constitutively expressed due to its potential role in bile resistance.

## 5. Conclusions

Together, these data show that ex-c-di-GMP inhibits the initial formation of biofilms and induces dispersion of formed biofilms of *C. jejuni*. It is possible to speculate that the bacterial cells that may not produce c-di-GMP, can nevertheless sense extraneously available molecules, such as QS molecules, during the pathogen-host–environment interactions. This study highlights the importance ex-c-di-GMP in regulating biofilm formation and dispersion by *C. jejuni* and indicates that the influence of external regulators in the life cycle of this organism, may need to be closely examined. Furthermore, the sensitivity of *C. jejuni* biofilms to ex-c-di-GMP makes them a promising drug target for controlling biofilm formation by using c-di-GMP analogues or similar small molecules [68].

## Figures and Tables

**Figure 1 microorganisms-10-02030-f001:**
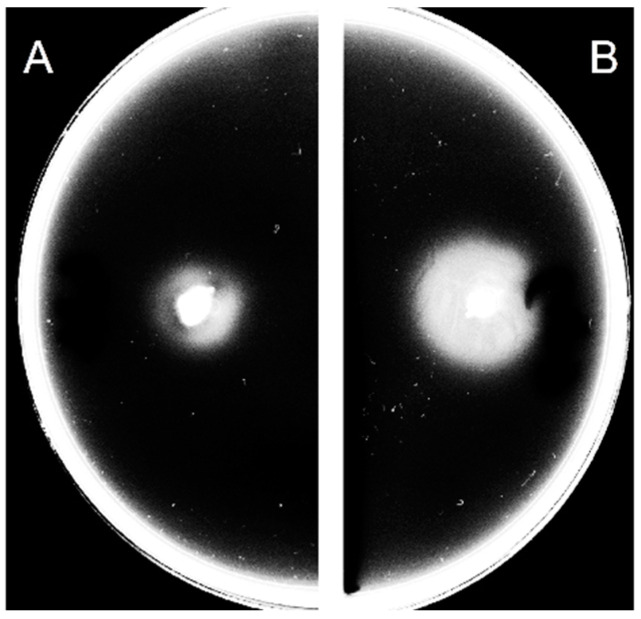
The swarming motility of *C. jejuni* 11168-O following 24 h incubation. (**A**) Wild type cells. (**B**) Wild type cells mixed with 200 µM of ex-c-di-GMP.

**Figure 2 microorganisms-10-02030-f002:**
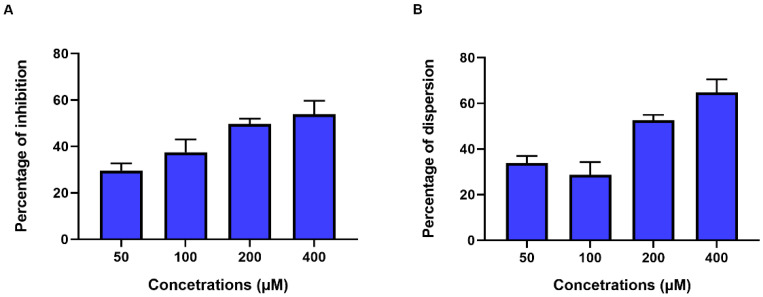
Inhibition and dispersion of *C. jejuni* 11168-O biofilms in the presence of ex-c-di-GMP at different concentrations. (**A**). Inhibition of biofilm formation by different concentrations of ex-c-di-GMP. (**B**) Dispersion of the existing biofilm induced by different concentrations of ex-c-di-GMP. Data are presented as mean ± standard error (3 or more biological repeats) of the percentage of inhibition (normalized to untreated control).

**Figure 3 microorganisms-10-02030-f003:**
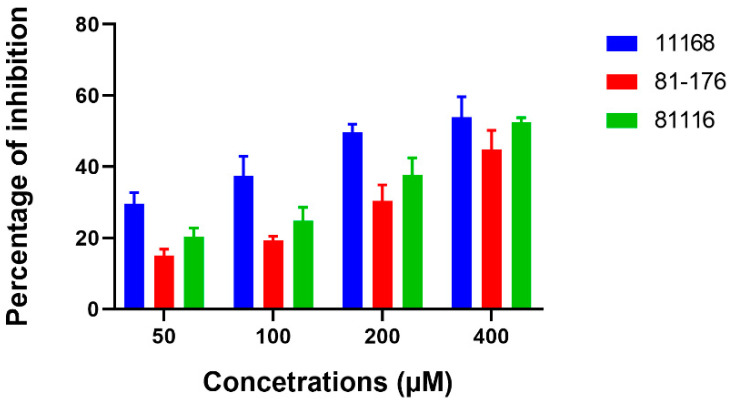
Inhibition of biofilm formation by different *C. jejuni* strains in the presence of 50–400 µM ex-c-di-GMP. *C. jejuni* 11168-O (11168), *C. jejuni* 81-176 (81-176), and *C. jejuni* 81116 (81116). Data are presented as mean ± standard error (3 or more biological repeats) of the percentage of inhibition (normalized to untreated control).

**Figure 4 microorganisms-10-02030-f004:**
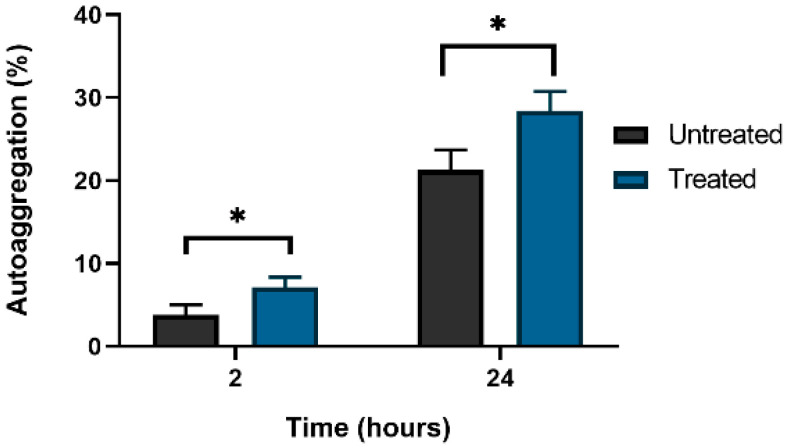
Percentage of *C. jejuni* auto-aggregation following ex-c-di-GMP treatment. Data are presented as mean ± standard errors (3 or more biological repeats). The asterisk (*) indicates a statistically significant difference compared to the untreated control (*p* < 0.05).

**Figure 5 microorganisms-10-02030-f005:**
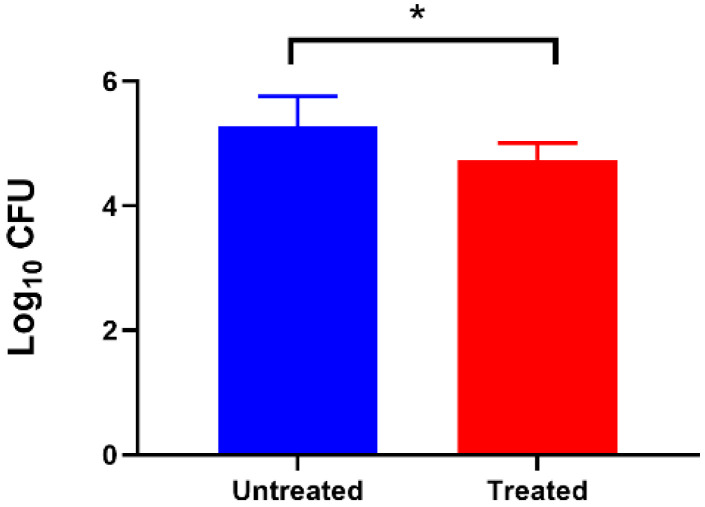
Effect of ex-c-di-GMP on adhesion of *C. jejuni* to Caco-2 cell. Inhibition of cell adhesion in the presence of 200 µM ex-c-di-GMP. Data are presented as mean ± standard error (3 or more biological repeats) and were analysed using two-tailed Student’s *t*-test. The asterisk (*) indicates a statistically significant difference compared to the untreated control (*p* < 0.05).

**Figure 6 microorganisms-10-02030-f006:**
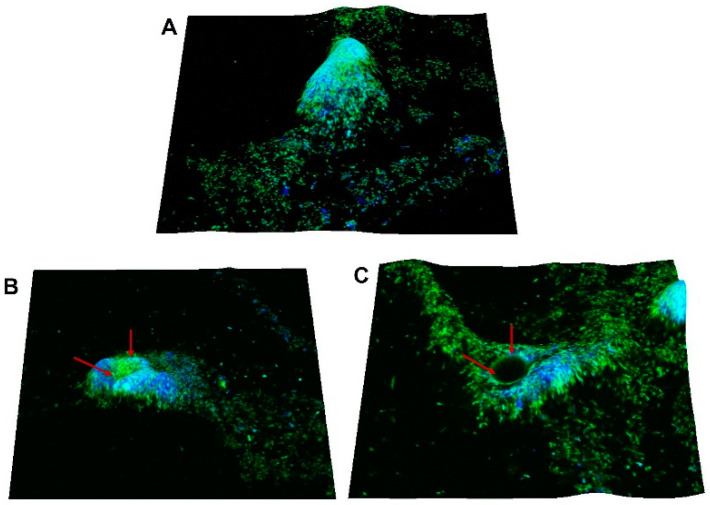
Void formation indicative of biofilm dispersion after exposure *C. jejuni* 11168-O biofilm to ex-c-di-GMP. (**A**) Representative CLSM images of biofilm grown for 48 h untreated (control). (**B**,**C**) Biofilm after exposure to 200 µM ex-c-di-GMP. Calcofluor White (CFW) (green, polysaccharides) and DAPI (blue, DNA). Red arrow indicates central hollowing of microcolony (Scale bar = 10 µm).

**Figure 7 microorganisms-10-02030-f007:**
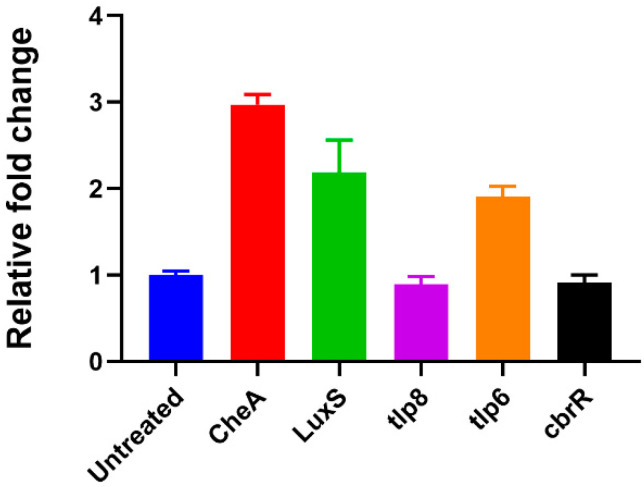
Analysis of the relative expression of genes in the presence of 200 µM ex-c-di-GMP. The expression of *tlp6*, *tlp8*, *luxS cheA* and *cbrR* genes relative to housekeeping gene *gryA* after incubation of *C. jejuni* 11168-O cells with 200 µM ex-c-di-GMP for 3 h.

## Data Availability

All the data is included in the article and the Appendix A.

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
