# Peer review of "Extracellular c-di-GMP Plays a Role in Biofilm Formation and Dispersion of Campylobacter jejuni"

_microorganisms, 2022, doi:10.3390/microorganisms10102030_

Round 1
Reviewer 1 Report
The manuscript entitled Extracellular c-di-GMP plays a role in biofilm formation and dispersion of Campylobacter jejuni presents information related to the inhibition of biofilm formation in C. jejuni strains by the action of c-di-GMP. The manuscript presents issues that authors must attend to prior to acceptance. It seems the manuscript was not revised by the authors. Below are the comments.
-Revise the writing of the manuscript. It was identified several typo errors.
-Separate units from values.
-Line 123. Change microaerobiclly by microaerobically.
-Section 2.7. Did the authors experiment with the adherence assay?
-What was the rationale for developing the RT-PCR analysis? The strains were obtained from laboratory collections, were they previously identified?
-Lines 189-193. The authors mentioned a significant effect (P<0.001) on the treatments shown in figure 2. What was the statistical analysis developed? The methods only mentioned the student t-test analysis.
-Figure 2. Indicate the significant differences by signaling the groups.
-Figure 3. The data must be analyzed by a means comparison test.
-Line 238. Change bioffilm by biofilm.
-Section 3.8. It was not described clearly the procedures for obtaining these results. The methodology only describes the RT-PCR procedures. Include details of the genes amplified, how were they constructed or where were they bought, etc.
Author Response
We are very much thankful to you for your time and interest to review this article. We have addressed your comments as following:
Reviewer1 comment 1. Revise the writing of the manuscript. It was identified several typo errors.
Author’s response: We have corrected the errors.
Reviewer1 comment 2. Separate units from values.
Author’s response: We have corrected that oversight.
Reviewer1 comment 3. Line 123. Change microaerobiclly by microaerobically.
Author’s response: We have corrected that oversight.
Reviewer1 comment 4. Section 2.7. Did the authors experiment with the adherence assay?
Author’s response: Yes, we did the adherence assay, and we’ve modified the section for clarity (Line 152-157).
Reviewer1 comment 5. What was the rationale for developing the RT-PCR analysis? The strains were obtained from laboratory collections, were they previously identified?
Author’s response: we have explained the rationale in the manuscript, now modified for clarity (Line 249-250). The strain was the original Skirrow isolate of NCTC C. jejuni 11168-O (Line 149). C. jejuni 81-176 (81-176), and C. jejuni 81116 (81116) are previously characterised human disease isolates. References had been included for clarity in the revised manuscript
Reviewer1 comment 6. Lines 189-193. The authors mentioned a significant effect (P<0.001) on the treatments shown in figure 2. What was the statistical analysis developed? The methods only mentioned the student t-test analysis.
Author’s response: The Student's t test was used to compare between two individual groups (untreated and treated ) to calculate the P values, however, we have calculated the percentage of biofilm inhibition (%) (Line 112-115 and Line 203-204) to facilitate the clarity of presentation and for consistency with many published articles using this method of data presentation (1, 2). For additional clarity, we have added Table S2 showing exact OD readings and analysis (supplementary data).
Reviewer1 comment 7. Figure 2. Indicate the significant differences by signaling the groups.
Author’s response: This figure is not showing whether there were significant differences between the groups, instead it is showing an increase in biofilm inhibition and dispersion in dose-dependent manner in presence of ex-c-di-GMP. We feel indication of significant differences is not appropriate for this figure.
Reviewer1 comment 8. Figure 3. The data must be analyzed by a means comparison test.
Author’s response: Please see comment 6 and 7
Reviewer1 comment 9. Line 238. Change bioffilm by biofilm.
Author’s response: We have corrected the error.
Reviewer1 comment 10. Section 3.8. It was not described clearly the procedures for obtaining these results. The methodology only describes the RT-PCR procedures. Include details of the genes amplified, how were they constructed or where were they bought, etc.
Author’s response: We have added more information regarding the gene primers in the methodology section (Lines 159-162).

Reviewer 2 Report
- The manuscript is exciting and discusses an important point of research. Some minor points to be considered:
- The manuscript should be revised for the English language and style.
- The introduction part should be shortened.
- Statistical analyses should be presented extensively in the results.
- The conclusion part should be rewritten to reveal the importance of the research findings.
Author Response
Reviewer 2 comment 1. The manuscript should be revised for the English language and style.
Author’s response: the manuscript has extensively revised for both language and style.
Reviewer 2 comment 2. The introduction part . should be shortened.
Author’s response: Reviewer has not mentioned which part of introduction should be shortened; however, we have made many changes to the introduction.
Reviewer 2 comment 3. Statistical analyses should be presented extensively in the results.
Author’s response: Statistical analyses has presented as recommended by the reviewer (see Reviewer1 comment 6).
Reviewer 2 comment 4. The conclusion part should be rewritten to reveal the importance of the research findings.
Author’s response: A statement highlighting the importance of our study had been added, lines 345-347.
We thank the Reviewers for their helpful comments and attention to detail, allowing us to prepare the best version of this manuscript we can
On behalf of all co-authors
Victoria Korolik.

Reviewer 3 Report
In this article, the authors studied the effect of extraneously added c-di-GMP on biofilm formation and dispersion of Campylobacter jejuni. The authors found that c-di-GMP has a profound effect on biofilm formation, motility, and the expression of quorum sensing and chemotaxis genes. In addition, the extraneous c-di-GMP inhibited adherence of C. jejuni to Caco-2 cells. The authors postulate that c-di-GMP could be considered as a target for developing new antimicrobial agents against C. jejuni. The manuscript is well written and nicely organized though there are some points to be addressed to improve the quality of the manuscript. Please find the comments below.
L17: Caco-2 cells, and upregulated
L27: causes bacterial
L78: strains, and it also
- Strain names in italics. Please check it throughout the manuscript
2.2: Chemicals used in this study
L96: Please re-write the sentence
2.4: The authors are not mentioning what chemical they tested and how much concentration they used?
2.5: Effect on C. jejuni growth. Effect of what? The title is not clear
The method is not written well. The authors should mention what they wanted to test and what concentrations they tested.
2.6: Please mention about treated and untreated samples and what concentration of the chemical was used.
2.7: There is no methodology written.
2.7: Adherence assay with cultured Caco-2 cells in vitro
- Please indicate statistical significance in each graph.
Figure 7 legend: 200 μM
Is there any study exploring biofilm formation, motility, and the expression of LuxS using c-di-GMP deletion mutant or an over-expression strain of C. jejuni? If yes, it would be interesting to discuss it in this paper.
L295: luxS mutation.
L295-296: Please re-write the sentence
L328-L330: ‘It is possible to speculate that the bacterial cells that may not produce c-di-GMP, can nevertheless sense extraneously available molecules, as QS molecules, during the pathogen-host–environment interactions.’ à Please provide clarification for this sentence.
L329: nevertheless; extraneously
- Please check spelling, spacing, and use of italics throughout the manuscript.
L334: Figure S2. The legend under Figure S2 in the supplementary file is better.

Author Response
Response to Reviewer’s comments:
In this article, the authors studied the effect of extraneously added c-di-GMP on biofilm formation and dispersion of Campylobacter jejuni. The authors found that c-di-GMP has a profound effect on biofilm formation, motility, and the expression of quorum sensing and chemotaxis genes. In addition, the extraneous c-di-GMP inhibited adherence of C. jejuni to Caco-2 cells. The authors postulate that c-di-GMP could be considered as a target for developing new antimicrobial agents against C. jejuni. The manuscript is well written and nicely organized though there are some points to be addressed to improve the quality of the manuscript. Please find te comments below.
Reviewer 3 comment 1. L17: Caco-2 cells, and upregulated
Author’s response: We have corrected the error.
Reviewer 3 comment 2. L27: causes bacterial
Author’s response: We have corrected the error.
Reviewer 3 comment 3. L78: strains, and it also
Author’s response: We have corrected the error.
Reviewer 3 comment 4. Strain names in italics. Please check it throughout the manuscript
Author’s response: We have corrected the error.
Reviewer 3 comment 5. 2.2: Chemicals used in this study
Author’s response: We have corrected the error.
Reviewer 3 comment 6. L96: Please re-write the sentence
Author’s response: the sentence has improved as recommended by the reviewer (Line 92-96)
Reviewer 3 comment 7. 2.4: The authors are not mentioning what chemical they tested and how much concentration they used?
Author’s response: this had been clarified.
Reviewer 3 comment 8. 2.5: Effect on C. jejuni growth. Effect of what? The title is not clear
Author’s response: this had been clarified.
Reviewer 3 comment 9. 2.5: The method is not written well. The authors should mention what they wanted to test and what concentrations they tested.
Author’s response: this had been clarified.
Reviewer 3 comment 10. 2.6: Please mention about treated and untreated samples and what concentration of the chemical was used.
Author’s response: this had been clarified.
Reviewer 3 comment 11. 2.7: There is no methodology written.
Author’s response: We have included methodology.
Reviewer 3 comment 12. 2.7: Adherence assay with cultured Caco-2 cells in vitro
Author’s response: this had been clarified.
Reviewer 3 comment 13. 2.7: Please indicate statistical significance in each graph.
Author’s response: this had been clarified.
Reviewer 3 comment 14. Figure 7 legend: 200 μM
Author’s response: We have corrected the error.
Reviewer 3 comment 15. Is there any study exploring biofilm formation, motility, and the expression of LuxS using c-di-GMP deletion mutant or an over-expression strain of C. jejuni? If yes, it would be interesting to discuss it in this paper.
Author’s response: No, there is no such study as far as we know, only one paper about c-di-GMP in C. jejuni DRH212 and we have discussed it (Line 322-330).
Reviewer 2 comment 16. L295: luxS mutation.
Author’s response: We have corrected the error.
Reviewer 3 comment 17. L295-296: Please re-write the sentence
Author’s response: the sentence has improved as recommended by the reviewer (Line 296-300).
Reviewer 3 comment 18. L328-L330: ‘It is possible to speculate that the bacterial cells that may not produce c-di-GMP, can nevertheless sense extraneously available molecules, as QS molecules, during the pathogen-host–environment interactions.’ à Please provide clarification for this sentence.
Author’s response: We have mentioned that c-di-GMP can be detected by the innate immune system (Line 321-329) and activate the expression of interferon genes and thus act as pathogen-associated molecular patterns (PAMPs) during infection. As c-di-GMP can be present in the environment, so the bacterial cells also could be sensing them, or it can enter the cells through unknown mechanisms (could be diffusion or transport). As we have shown, LuxS is upregulated after treatment with ex-c-di-GMP. LuxS/AI-2 QS system acts as a global regulatory mechanism for the cellular processes include the motility, biofilm formation, and drug resistance (3-5). This suggests the presence of regulatory connections (direct or indirect) between the c-di-GMP and quorum sensing system, LuxS, in C. jejuni 11168. For example, in Pseudomonas aeruginosa, the reduction of intracellular c-di-GMP increased the gene expression of quorum sensing (6,7). However, the molecular bases for ex c-di-GMP inhibition of C. jejuni biofilm is unknown.
Reviewer 3 comment 19. L329: nevertheless; extraneously
Author’s response: We have corrected that.
Reviewer 3 comment 20. Please check spelling, spacing, and use of italics throughout the manuscript.
Author’s response: We have checked and corrected that.
Reviewer 3 comment 21. L334: Figure S2. The legend under Figure S2 in the supplementary file is better.
Author’s response: We have corrected that.
We thank the Reviewers for their helpful comments and attention to detail, allowing us to prepare the best version of this manuscript we can
On behalf of all co-authors
Victoria Korolik.

Round 2
Reviewer 1 Report
The manuscript has been improved. The authors have attended to all the comments. I recommend accepting the manuscript.